# Effects of aging and risk-taking behaviors on fatal injuries among old motorcyclists in Taiwan: Evidence from 2011 to 2022

Akhmad Fajri Widodo[1], Cheng-Wei Chan[1,2,3,4], Hon-Ping Ma[1,5,6], Chen-Chun Shu[7], Julie Brown[7,8], Chung-Jen Chao[9], Shou-Chien Hsu[4,10], Shih Yu Ko[6], Hui-An Lin[1,11,5], Chenyi Chen[1], Chun-Chieh Chao[1,11,5], Iftitakhur Rohmah[12], Tung-Yao Tsai[5,6], Chih-Wei Pai[1]*

1 Graduate Institute of Injury Prevention and Control, College of Public Health, Taipei Medical University, Taipei City, Taiwan, 2 Department of Emergency Medicine, New Taipei City Hospital, New Taipei City, Taiwan, 3 College of Medicine, Chang Gung University, Taoyuan City, Taiwan, 4 Department of Emergency Medicine, Chang Gung Memorial Hospital, Linkou Branch, Taoyuan, Taiwan, 5 Department of Emergency Medicine, School of Medicine, College of Medicine, Taipei Medical University, Taipei, Taiwan, 6 Department of Emergency Medicine, Taipei Medical University-Shuang Ho Hospital, New Taipei City, Taiwan, 7 The George Institute for Global Health, University of New South Wales, Sydney, New South Wales, Australia, 8 Neuroscience Research Australia, Sydney, New South Wales, Australia, 9 Department of Traffic Science, Central Police University, Taoyuan City, Taiwan, 10 Department of Occupational Medicine, Chang Gung Memorial Hospital, Linkou Branch, Taoyuan, Taiwan, 11 Department of Emergency Medicine, Taipei Medical University Hospital, Taipei City, Taiwan, 12 School of Nursing, College of Nursing, Taipei Medical University, Taipei, Taiwan

☯ These authors contributed equally to this work.
* cpai@tmu.edu.tw

## Abstract

### Background and objectives

Fatality rates of motor vehicle crashes among the old population have risen, primarily in association with age-related declines in health and functional abilities. Comparatively little research has been conducted to examine the impacts of risk-taking behaviors (such as unhelmeted, unlicensed, and drunk riding) on fatalities among old motorcyclists.

### Materials and methods

This study employed the Taiwan National Traffic Crash Dataset from 2011 to 2022 to investigate fatal injuries among old motorcyclists. To identify risk factors associated with injury fatality, chi-square tests were employed. Subsequently, stepwise logistic regression models incorporating multiple variables were constructed to examine these factors. To explore potential interactions between aging and specific behavioral variables, joint effect analysis was performed, followed by a subgroup analysis specific to very old motorcyclists (75 years or above).

**Data availability statement:** The police-reported crash data, which are open to researchers in Taiwan, are available from the Health and Welfare Data Science Center (http://dep.mohw.gov.tw/DOS/np-2497-113.html). Only citizens of Taiwan who fulfill the requirements of conducting research projects are eligible to apply for the police-reported crash dataset. The use of police-reported crash dataset is limited to research purposes only. Applicants must follow the Computer-Processed Personal Data Protection Law. To support transparency and reproducibility, we have made available the minimal anonymized dataset used in our current study, which has been de-identified in accordance with data-sharing guidelines and can be accessed via the following link: https://figshare.com/articles/dataset/data/29318324/1?file=55377464.

**Funding:** This study was financially supported by grants from the National Science and Technology Council, Taiwan (NSTC 112-2410-H-038-023-MY2; NSTC 110-2410-H-038-016-MY2) and New Taipei City Hospital (NTPC 113-002). The funders had no role in the design of the study, data collection and analysis, interpretation of data, or preparation of the manuscript.

**Competing interests:** The authors have declared that no competing interests exist.

**Abbreviations:** AOR, adjusted odds ratio; CI, confidence interval; SD, standard deviation.

## Results

The analyzed dataset comprised 338,231 crashes involving old motorcyclists, of which 2,703 were fatal and 335,528 were nonfatal. A linear relationship was observed between the risk of fatal injuries and increasing age. The adverse effect of aging on fatalities was exacerbated among unhelmeted (AOR = 4.46; CI = 3.79–5.26), unlicensed (AOR = 1.76; CI = 1.55–1.99), and drunk riders (AOR = 5.35; CI = 4.06–7.06). Our subgroup analysis revealed similar detrimental effects of these three behaviors among very old motorcycles when they were unhelmeted (AOR = 3.67; CI = 3.09–4.36), unlicensed (AOR = 1.31; CI = 1.13–1.50) and intoxicated (AOR = 3.06; CI = 2.31–4.06). Further stratification by geographic location demonstrated that certain risk-taking behaviors, such as unhelmeted and drunk riding, were consistently associated with higher fatality rates. Notably, the magnitude of the effects of drunk riding in rural settings on fatal injuries was higher.

## Conclusions

The study established a linear relationship between age and the risk of fatal injuries among old motorcyclists. The three risk-taking behaviors apparently are risk factors for fatal injuries, and evident when motorcyclists were very old. This underscores the importance of tailored interventions targeting individuals aged 75 years or above who engage in unlicensed riding, which remains an important area for continued research.

## Introduction

The risk of motor vehicle crashes and fatalities increases with age [1]. Studies have established an association between older age, diminished health-related functional abilities, and impaired riding performance, culminating in an increased crash risk among old drivers and riders, even when considering the wide range of health conditions and riding experience within the old population [2,3]. These aging effects were linked to other factors, such as risk-taking behavior, which led to a higher likelihood of sustaining fatalities among old motorcyclists [4]. Despite their age-related conditions and risk-taking behaviors, old motorcyclists continue to ride motorcycles potentially exacerbating crash risks and increasing the likelihood of fatal injuries [4,5].

The number of motor vehicle crashes and fatalities involving old individuals and other road users increased from 7,902 in 2020–9,102 in 2021 [6,7]. Old riders exhibited a higher propensity for involvement in single-motor vehicle crashes, sideswipe crashes, angle crashes, rear-end crashes, head-on crashes, intersection crashes, and crossroad crashes [8–10]. These crashes were more likely to occur at unsignalized intersections [11] and were attributed to factors such as fatigue, illness, and unintended acceleration [12,13].

The number of fatalities due to motorcycle crashes is higher than the number of deaths due to other motor vehicle crashes [8]. Motorcyclists constituted a substantial 78% of all road users involved in motor vehicle crashes [14], with old motorcyclists

comprising 23% of these crashes [15]. Notably, the Ministry of Transportation and Communications in Taiwan reported that approximately 40% of all motorcycle crashes involved old riders, of whom 75% sustained fatal injuries [14]. Comparable trends were observed in the United States and Thailand, where old motorcyclists exhibited an increased risk of fatal injuries compared with their younger counterparts [9,16].

Although previous research has predominantly focused on motorcycle injuries in general, research on the effect of age on fatal injuries among old motorcyclists is scarce. Additionally, the association between risk-taking behaviors (such as unhelmeted, unlicensed, and drunk riding) and fatal injuries among this population has not been thoroughly examined. This study addressed this research gap by investigating the relationship between age and fatal injuries among old motorcyclists and examined the effects of several risk-taking behaviors on fatal injuries.

## Materials and methods

### Data source

The present study analyzed fatalities sustained by old motorcyclists by using the Taiwan National Traffic Crash Dataset. The dataset, covering the period from 01/01/2011–31/12/2022, were accessed for the research purpose on 11/11/2024. This dataset is managed by the National Police Agency and comprises all reported road traffic crashes. Qualified and experienced police crash investigators compile three distinct files, specifically crashes, vehicle, and victim files. Crash files contain general information such as crash date, time, weather conditions, and road type. Vehicle files contain information about vehicle characteristics, and victim files contain information about casualty data, including age, sex, injury fatality, injured body regions, protective equipment use, license status, and alcohol use. In our dataset, detailed clinical information such as Injury Severity Scores (ISS) or Abbreviated Injury Scale (AIS) are not available. Therefore, injury fatality is categorized as fatal or nonfatal. Fatal injury was defined as a case in which the riders died following crashes, whereas a non-fatal injury included any other recorded injury category aside from death. To ensure accuracy, crash investigators collect injury data from hospitals within a 30-day period, documenting primary diagnosis, injured body regions, and injury fatality.

Fig 1 presents a flowchart of sample selection from the Taiwan National Traffic Crash Dataset for the period spanning from 2011–2022. An initial pool of 770,678 old casualties was identified. Due to missing data on sex, injury fatality, and road user types, 23,335 samples were excluded. Furthermore, 409,112 records were excluded because they involved nonmotorcyclists, nonfatal injuries, or unknown independent variable categories. The final dataset comprised 338,231 old motorcyclist casualties, categorized into 2,703 fatal and 335,528 nonfatal injuries. The data that support the findings of this study are openly available at https://figshare.com/s/385dde866b4a8964cef1.

### Definitions of variables

Demographic data collected for casualties included age (categorized into five groups: 65–69, 70–74, 75–79, 80–84, and ≥85 years), sex (male or female), helmet use (helmeted or unhelmeted), license status (licensed or unlicensed), and alcohol use (drunk or not drunk). Alcohol use was determined through police-administered breathalyzer tests or subsequent blood tests at hospitals. In Taiwan, riders with breathalyzer test results of ≥0.15 mL/L or blood alcohol concentrations of >0.03% are considered to be drunk. Mandatory breathalyzer tests for motorcyclists involved in crashes provided alcohol use data for this study. Information on injured body regions encompassed included head and neck, upper or lower extremities, chest or abdomen, spine, and unknown categories when the injury body parts were not reported.

Two temporal factors were examined: day of the crash (weekday or weekend) and time of crash, categorized as rush hour (07:00–08:59 and 17:00–18:59), non–rush hour (09:00–16:59), evening (19:00–23:59), and midnight/ early morning (00:00–06:59). The following factors were considered: crash location (rural: roadways with speed limits of ≥51 km/h; urban: roadways with speed limits of ≤50 km/h), crash type (head-on, crossroad, single motorcycle,

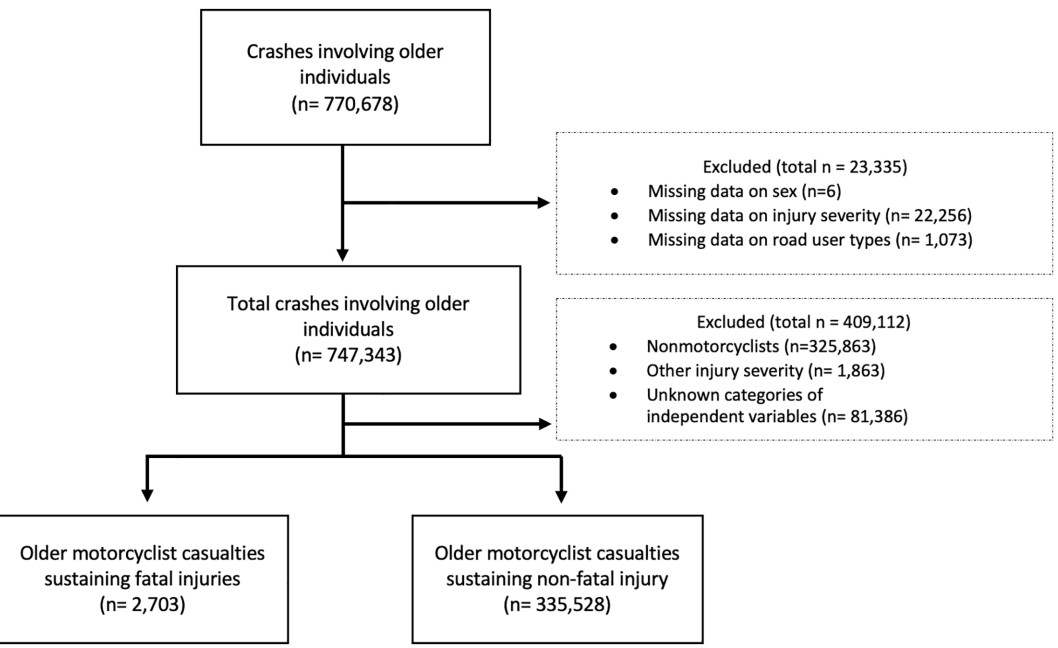

**Fig 1. Sample selection flowchart.**

motorcycle-pedestrian, and other crashes (including sideswipe, rear-end, T-Bone, and unknown crash types), weather condition (fine or adverse weather), light conditions (daylight, lit or unlit), road surface condition (dry or slippery), and sight distance (adequate or limited due to obstructions such as road curvatures, buildings, and trees).

## Statistical analysis

The distribution of motorcyclist injury fatality across various independent variables was examined (Table 1). Chi-square tests were employed to identify significant associations ($p < 0.05$) between these variables and injury fatality in old motorcyclists and to identify variables significantly associated with the outcome variables. Subsequently, stepwise logistic regression models were used to estimate the odds of fatal injuries after controlling for specific variables.

According to Ministry of Transportation and Communications (MOTC) Taiwan, mandatory health examination is required for license renewal for Taiwanese individuals aged 75 years old or above [17]; therefore, an age of 75 years was selected as a cutoff to analyze fatality rate variations across different variables. Joint effect analysis was performed to examine whether very old motorcyclists (75 years or above) engaging in several risk-taking behaviors (such as unhelmeted, unlicensed, and drunk riding) exhibited a higher risk of fatal injuries compared to their younger counterparts. This model incorporates the interaction effects of very old motorcyclists and younger counterparts (i.e., 75 years old or above vs. younger ones) as well as the risk-taking behaviors, yielding two x two combinations: e.g., 75 years old or above engaging in unlicensed riding, 75 years old or above engaging in licensed riding, younger motorcyclists engaging in unlicensed riding, younger motorcyclists engaging in licensed riding. The category with the highest adjusted odds ratio takes on the value 1 and the other three categories are combined and take the value 0 (otherwise).

To illustrate the effectiveness of models with joint effects, we found that these models produced a higher log-likelihood at convergence and demonstrated an improved overall fit, as indicated by a better $\rho^2$ statistic.

Moreover, we performed a likelihood ratio test (e.g., e) to confirm the superiority of the joint effects models over the general models. The test statistic is given by:

**Table 1. Factors associated with injury fatality among old motorcyclists.**

| Variable | Fatal | Nonfatal | Total | $\chi^s$ test |
|---|---|---|---|---|
| | (n = 2,703) | (n = 335,528) | (n = 338,231) | p value |
| **Age (years)**, n (%) | | | | <0.001 |
| 65-69 | 754 (0.5%) | 142,019 (99.5%) | 142,773 (42.2%) | |
| 70-74 | 725 (0.8%) | 95,085 (99.2%) | 95,810 (28.3%) | |
| 75-79 | 645 (1.1%) | 57,859 (98.9%) | 58,504 (17.3%) | |
| 80-84 | 389 (1.3%) | 29,276 (98.7%) | 29,665 (8.8%) | |
| ≥85 | 190 (1.7%) | 11,289 (98.3%) | 11,479 (3.4%) | |
| **Sex**, n (%) | | | | <0.001 |
| Male | 1,936 (0.9%) | 206,581 (99.1%) | 208,517 (61.6%) | |
| Female | 767 (0.6%) | 128,947 (99.4%) | 129,714 (38.4%) | |
| **Helmet use**, n (%) | | | | <0.001 |
| Helmeted | 2,324 (0.7%) | 327,890 (99.3%) | 330,214 (97.6%) | |
| Unhelmeted | 379 (4.7%) | 7,638 (95.3%) | 8,017 (2.4%) | |
| **License status**, n (%) | | | | <0.001 |
| Licensed | 2,008 (0.7%) | 272,465 (99.3%) | 274,473 (81.1%) | |
| Unlicensed | 695 (1.1%) | 63,063 (98.9%) | 63,758 (18.9%) | |
| **Alcohol use**, n (%) | | | | <0.001 |
| Not drunk | 2,515 (0.8%) | 331,656 (99.2%) | 334,171 (98.8%) | |
| Drunk | 188 (4.6%) | 3,872 (95.4%) | 4,060 (1.2%) | |
| **Speed limit**, n (%) | | | | <0.001 |
| Urban (≤50 km/h) | 2,270 (0.7%) | 315,792 (99.3%) | 318,062 (94%) | |
| Rural (≥51 km/h) | 433 (2.1%) | 19,736 (97.9%) | 20,169 (6%) | |
| **Crash time**, n (%) | | | | <0.001 |
| Midnight/early morning (0–6) | 163 (1.6%) | 9,740 (98.4%) | 9,903 (2.9%) | |
| Rush hours (7–8/17–18) | 872 (0.8%) | 111,578 (99.2%) | 112,450 (33.2%) | |
| Nonrush hours (9–16) | 1,544 (0.8%) | 193,667 (99.2%) | 195,211 (57.7%) | |
| Evening (19–23) | 124 (0.6%) | 20,543 (99.4%) | 20,667 (6.1%) | |
| **Light condition**, n (%) | | | | <0.001 |
| Daylight | 2,312 (0.8%) | 288,511 (99.2%) | 290,823 (86%) | |
| Morning/twilight | 100 (1.1%) | 9,060 (98.9%) | 9,160 (2.7%) | |
| Darkness-lit | 261 (0.7%) | 36,253 (99.3%) | 36,514 (10.8%) | |
| Darkness-unlit | 30 (1.7%) | 1,704 (98.3%) | 1,734 (0.5%) | |
| **Day of crashes**, n (%) | | | | 0.468 |
| Weekend | 551 (0.8%) | 69,493 (99.2%) | 70,044 (20.7%) | |
| Weekday | 1,862 (0.8%) | 227,851 (99.2%) | 229,713 (67.9%) | |
| Public holiday | 290 (0.8%) | 38,184 (99.2%) | 38,474 (11.4%) | |
| **Crash type**, n (%) | | | | <0.001 |
| Head-on crashes | 100 (0.7%) | 14,108 (99.3%) | 14,208 (4.2%) | |
| Crossroad crashes | 608 (1.3%) | 45,756 (98.7%) | 46,364 (13.7%) | |
| Single-motorcycle crashes | 353 (1.5%) | 22,789 (98.5%) | 23,142 (6.8%) | |
| Motorcycle-pedestrian crashes | 14 (0.3%) | 5,132 (99.7%) | 5,146 (1.5%) | |
| Otherwise | 1,628 (0.7%) | 247,743 (99.3%) | 249,371 (73.7%) | |
| **Sight distance**, n (%) | | | | <0.001 |
| Adequate | 2,630 (0.8) | 329,821 (99.2%) | 332,451 (98.3%) | |
| Limited | 73 (1.3%) | 5,707 (98.3%) | 5,780 (1.7%) | |

*(Continued)*

**Table 1.** (Continued)

| Variable | Fatal (n = 2,703) | Nonfatal (n = 335,528) | Total (n = 338,231) | $\chi^s$ test p value |
|---|---|---|---|---|
| **Injury part**, n (%) | | | | <0.001 |
| Head & neck | 951 (8.6%) | 10,077 (91.4%) | 11,028 (3.3%) | |
| Extremities | 56 (0.1%) | 68,479 (99.9%) | 68,535 (20.3%) | |
| Chest/abdomen | 129 (3.2%) | 3,925 (96.8%) | 4,054 (1.2%) | |
| Spine | 2 (0.1%) | 1,621 (99.9%) | 1,623 (0.5%) | |
| Unknown | 1,565 (0.6%) | 251,426 (99.4%) | 252,991 (74.8%) | |
| **Road surface condition**, n (%) | | | | 0.001 |
| Dry | 2,534 (0.8%) | 308,516 (99.2%) | 311,050 (92%) | |
| Slippery | 169 (0.6%) | 27,012 (99.4%) | 27,181 (8%) | |
| **Weather**, n (%) | | | | 0.243 |
| Adverse | 353 (0.8%) | 46,433 (99.2%) | 46,786 (13.8%) | |
| Fine | 2,350 (0.8%) | 289,095 (99.2%) | 291,445 (86.2%) | |

$$\chi^2 = -2[LL(\beta_G) - LL(\beta_J)]$$

where $LL(\beta_G)$ represents the log-likelihood at convergence for the general model, and $LL(\beta_J)$ is for the joint effects model [18]. This statistic follows a $\chi^2$ distribution, with degrees of freedom equal to the difference in the number of parameters between the general and joint effects models.

In addition, a subgroup analysis was conducted specifically for very old motorcyclists who are required to undergo a health examination. This particular group of motorcyclists is prohibited from riding if they are not fit enough. The aim of this subgroup analysis was to examine whether those three risk-taking behaviors (i.e., unhelmeted, unlicensed, and drunk riding) still play a role in their injuries. Furthermore, the analysis was stratified by geographic locations such as urban and rural settings.

## Ethics approval and consent to participate

The current research analyzed national crash data without individuals' confidential information such as names or identity numbers. In addition, this study was conducted in accordance with the Declaration of Helsinki principles and approved by the Research Ethics Review Committee, New Taipei City Hospital (113001-E) on March 4th, 2024. Therefore, the requirement for informed consent was waived and consent to participate was not requested.

## Results

### Motorcyclist fatalities

A total of 338,231 old motorcyclist casualties were identified. Among these, 2,703 were fatal (0.8%), and 335,528 were nonfatal (99.2%). The 65–69 age group constituted the largest proportion of casualties at 142,773 (42.2%). Although only 11,479 (3.4%) of the casualties were aged 85 years or above, their fatality rate was the highest among all age groups (190; 1.7%). Male old motorcyclists exhibited a higher fatality rate (1,936; 0.9%) compared with female old motorcyclists (767; 0.6%). Risk factors associated with increased fatality rates included unhelmeted riding (379; 4.7%), unlicensed riding (695; 1.1%), and drunk riding (188; 4.6%). The majority of motorcycle crashes involving old riders were in urban settings (318,062; 94%), during non–rush hour (09:00–16:59; 195,211; 57.7%), in daylight conditions (290,823; 86%), on weekdays (229,713; 67.9%), with adequate sight distance (332,451; 98.3%), on dry road surfaces (311,050; 92%), and during fine weather (291,445; 86.2%).

Chi-squared tests revealed significant associations between the outcome variable and age, sex, helmet use, license status, alcohol consumption, speed limit, crash time, light conditions, crash type, sight distance, injury part, and road surface conditions. These variables were subsequently entered into stepwise logistic regression models.

Table 2 presents the results of stepwise logistic regression analyses. A linear relationship was observed between the risk of fatal injuries and increasing age. Compared with those aged 65–69 years, motorcyclists in the age groups 70–74, 75–79, 70–84, and ≥85 years exhibited 1.299, 1.700, 1.963, and 2.430 times higher odds of fatal injuries, respectively (adjusted odds ratios [AORs] = 1.299, 1.700, 1.963, and 2.430; confidence intervals [CIs] = 1.170–1.442, 1.524–1.897, 1.725–2.235, and 2.051–2.879). Additional risk factors for fatal injuries included risk-taking behaviors such as unhelmeted riding (AOR = 3.581; CI = 3.171–4.040), unlicensed riding (AOR = 1.382; CI = 1.259–1.517), and drunk riding (AOR = 3.861; CI = 3.266–4.564).

### Joint effects of age and risk-taking behaviors on old motorcyclist fatalities

Fig 2 displays a forest plot illustrating the interactive effects of age and risk-taking behaviors on fatal injuries among old motorcyclists. The results indicated a synergistic relationship between very old motorcyclists and the following factors: unhelmeted riding (AOR = 4.46, CI = 3.79–5.26), unlicensed riding (AOR = 1.76, CI = 1.55–1.99), and drunk riding (AOR = 5.35, CI = 4.06–7.06) compared with otherwise.

### Subgroup analysis among very old motorcyclists in urban and rural settings

Fig 3 displays a forest plot depicting the interaction between age and risk-taking behaviors on fatal injuries among very old motorcyclists. The findings revealed that the likelihood of fatal injuries was significantly higher among very old motorcyclists who were unhelmeted (AOR = 3.67, CI = 3.09–4.36), unlicensed (AOR = 1.31, CI = 1.13–1.50), and drunk riding (AOR = 3.06, CI = 2.31–4.06). In addition, the results indicated that in urban settings, very old motorcyclists were more likely to sustain fatal injuries when they were unhelmeted (AOR = 3.65, CI = 3.03–4.40), unlicensed (AOR = 1.33, CI = 1.14–1.55), and drunk riding (AOR = 2.80, CI = 2.03–3.86). In rural settings, unhelmeted riding similarly increased the risk of fatal injuries (AOR = 3.65, CI = 3.34–5.71), while the magnitude of the effect of drunk riding was larger (AOR = 4.35, CI = 2.39–7.94).

## Discussion

Our analysis of the Taiwan National Crash Data database revealed a linear correlation between the risk of fatal injuries and the increasing age among old motorcyclists. This finding aligns with previous research attributing the heightened vulnerability of old riders to decreased physical resilience, slower reaction times, and an increased prevalence of comorbidities [15,19–22]. To mitigate the risk of crashes and fatal injuries, several countries, including Taiwan, Japan, Singapore, Hong Kong, and the United Kingdom, have implemented mandatory health examinations for old riders [17]. Studies have demonstrated that such health examinations, particularly vision tests, can effectively reduce fatal crashes among old riders [23–25]. By identifying and addressing underlying health issues, these examinations have the potential to improve overall health outcomes, reduce health-care costs associated with severe injuries [26,27], and ultimately prevent motorcycle-related fatalities among old riders [22,28].

Our findings elucidated that those risk-taking behaviors, including unhelmeted riding, unlicensed riding, and drunk riding, were associated with fatal injuries. In addition, unhelmeted and intoxicated riding were associated with higher fatality rates when stratified by geographic location. Specifically, in rural settings, the magnitude of the effect of intoxicated riding on fatal injuries was larger. These findings are consistent with one previous study concluding that the likelihood of fatal injuries among old motorcyclists increased significantly when they were unhelmeted [29]. A possible explanation is that old motorcyclists are physiologically more vulnerable to sustain severe injuries owing to factors such as decreased bone density and slower healing [30–32]. Moreover, unhelmeted riding was associated with higher fatality rates among old

**Table 2. Odds of fatal injuries sustained by old motorcyclists.**

| Variable | AOR | 95% CI | p value |
|---|---|---|---|
| **Age (years)** | | | |
| 65-69 | Ref | | |
| 70-74 | 1.299 | 1.170 - 1.442 | <0.001 |
| 75-79 | 1.700 | 1.524 - 1.897 | <0.001 |
| 80-84 | 1.963 | 1.725 - 2.235 | <0.001 |
| ≥85 | 2.430 | 2.051 - 2.879 | <0.001 |
| **Sex** | | | |
| Male | 1.359 | 1.239–1.490 | <0.001 |
| Female | Ref | | |
| **Helmet use** | | | |
| Helmeted | Ref | | |
| Unhelmeted | 3.581 | 3.171–4.040 | <0.001 |
| **License status** | | | |
| Licensed | Ref | | |
| Unlicensed | 1.382 | 1.259–1.517 | <0.001 |
| **Alcohol use** | | | |
| Not drunk | Ref | | |
| Drunk | 3.861 | 3.266–4.564 | <0.001 |
| **Speed limit** | | | |
| Urban (≤50 km/h) | Ref | | |
| Rural (≥51 km/h) | 2.423 | 2.173–2.701 | <0.001 |
| **Crash time** | | | |
| Midnight/early morning (0–6) | 2.209 | 1.697–2.874 | <0.001 |
| Rush hours (7–8/17–18) | 1.224 | 0.968–1.548 | 0.091 |
| Nonrush hours (9–16) | 1.269 | 0.995–1.620 | 0.055 |
| Evening (19–23) | Ref | | |
| **Light condition** | | | |
| Daylight | Ref | | |
| Morning/twilight | 1.164 | 0.937–1.445 | 0.170 |
| Darkness-lit | 1.030 | 0.866–1.226 | 0.736 |
| Darkness-unlit | 1.609 | 1.081–2.395 | 0.019 |
| **Crash type** | | | |
| Head-on crashes | 1.079 | 0.881–1.321 | 0.464 |
| Crossroad crashes | 2.022 | 1.841–2.221 | <0.001 |
| Single-motorcycle crashes | 2.357 | 2.099–2.647 | <0.001 |
| Motorcycle-pedestrian crashes | 0.415 | 0.245–0.703 | 0.001 |
| Otherwise | Ref | | |
| **Sight distance** | | | |
| Adequate | Ref | | |
| Limited | 1.276 | 0.999–1.630 | 0.051 |
| **Injury part** | | | |
| Head & neck | 12.729 | 11.677–13.876 | <0.001 |
| Otherwise | Ref | | |
| **Weather** | | | |
| Adverse | 0.897 | 0.799–1.008 | 0.068 |
| Fine | Ref | | |

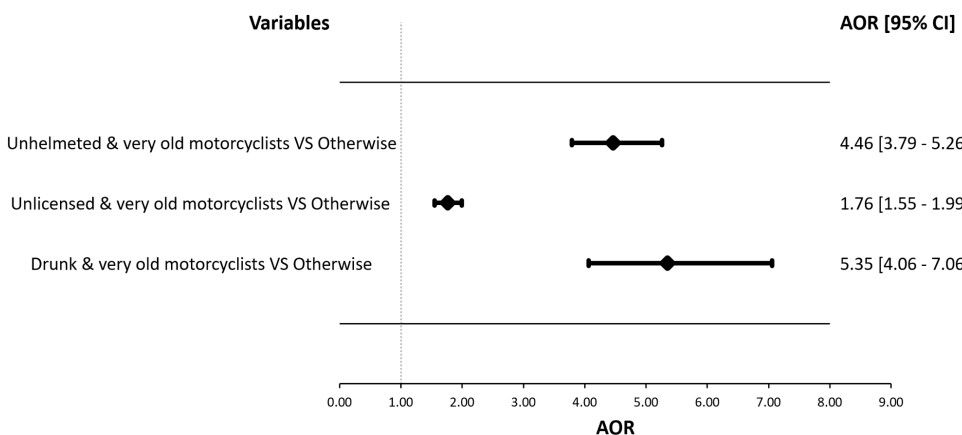

**Fig 2. Joint effects of age and risk-taking behaviors on old motorcyclist fatalities (n = 419,617).**

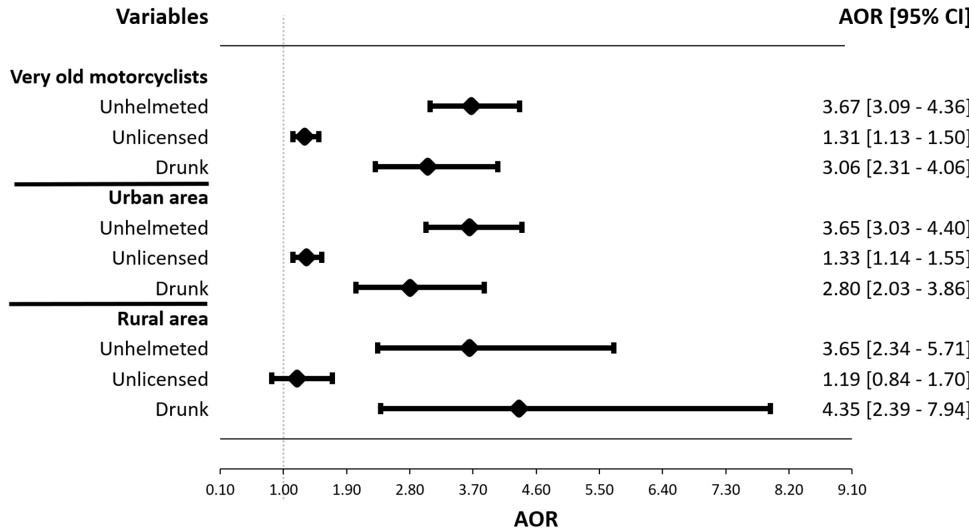

**Fig 3. Subgroup analysis among very old motorcyclists in urban and rural settings (n = 124,897).**

motorcyclists, possibly attributable to diminished resilience and decline in physical fitness [28,33–35]. Therefore, given the proven effectiveness of helmet use in reducing fatalities [36–39], targeted interventions focusing on old motorcyclists are warranted.

Our results also revealed that old motorcyclists were more likely to sustain fatal injuries when they were unlicensed. This result aligns with the previous studies, which have identified unlicensed riding as risk factor for fatal injuries [40–42]. The absence of a valid license also often exhibited noncompliance with road safety regulations, significantly increasing their risks of fatal injuries [43,44]. Hence, unlicensed old motorcyclists represent a critical public health concern that deserves further discussion.

In response to improving old motorcyclists' safety, several laws have been implemented [17,45]. In Taiwan, very old motorcyclists are mandated to undergo health examination to renew their license [17]. Similarly, this law appeared to be associated with a decrease in risks of fatal injuries when implemented in Japan [45]. These findings emphasized the

beneficial effects of health examinations for license renewal among old motorcyclists. However, the risk of fatal injuries may be increased among old motorcyclists who fail to renew their license and continue to ride. Therefore, governments are encouraged to facilitate and promote the use of public transportation, particularly among old population, to enhance road safety and mobility.

Consistent with the previous studies [46–50], our results indicated that the magnitude of intoxicated-riding effect on fatal injuries in rural settings was 4.35 times larger. It is possible that emergency response to alcohol-related crashes that occur in rural areas is slower than in other conditions, exacerbating unattended injuries in particular among old motorcyclists [51]. Past research suggested that this may also be attributed to the fact that intoxicated riding increased the likelihood of speeding, particularly in rural areas, which significantly increased the risk of fatal injuries [52]. Therefore, stricter enforcement of drunk riding laws, such as license revocation was needed in reducing fatal injuries [53,54]. Moreover, additional sanction such as increased the maximum jail terms of drunk riding should be implemented to improve the road users' safety [55].

Our study has several limitations due to data constraints within the police-reported crash dataset. Specifically, information regarding crash partners, such as their age, vehicle type, and behavior, is missing, potentially influencing the study's findings. These characteristics are important, as they can significantly affect crash outcomes. For instance, motorcycle riders were more likely to sustain fatal injuries when they were struck by unlicensed, unrestrained, and drunk car drivers [29,47]. Additionally, although this study focused on specific variables, the effects of comorbidities on crash involvement and injury fatality among old motorcyclists remain an important area for future research. Evaluating the effectiveness of health examination laws for old motorcyclists represent a promising avenue for future investigations.

## Conclusion

This study established a linear correlation between increasing age and an elevated risk of fatal injuries among old motorcyclists. These individuals demonstrate significantly elevated fatality risks, particularly when engaging in risk-taking behaviors such as unhelmeted riding, unlicensed riding, and drunk riding particularly for those aged 75 years or above. Focused enforcement measures and educational programs aimed at old motorcyclists are essential to reduce these risks. Tailored interventions among individuals with a combination of 75 years or above and unlicensed riding are an area of future research.

## Acknowledgments

This manuscript was edited by Wallace Academic Editing.

## Author contributions

**Data curation:** Akhmad Fajri Widodo, Hui-An Lin, Chun-Chieh Chao, Iftitakhur Rohmah.

**Formal analysis:** Akhmad Fajri Widodo, Hui-An Lin, Chun-Chieh Chao, Iftitakhur Rohmah.

**Funding acquisition:** Cheng-Wei Chan.

**Methodology:** Shih Yu Ko, Chenyi Chen, Chun-Chieh Chao.

**Supervision:** Chen-Chun Shu, Julie Brown, Tung-Yao Tsai, Chih-Wei Pai.

**Validation:** Akhmad Fajri Widodo, Chung-Jen Chao, Shou-Chien Hsu.

**Writing – original draft:** Akhmad Fajri Widodo.

**Writing – review & editing:** Akhmad Fajri Widodo, Cheng-Wei Chan, Hon-Ping Ma.

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
