## [Decision Letter · Decision Letter 0]

15 May 2025

Dear Dr. Pai,

Thank you for submitting your manuscript to PLOS ONE. After careful consideration, we feel that it has merit but does not fully meet PLOS ONE’s publication criteria as it currently stands. Therefore, we invite you to submit a revised version of the manuscript that addresses the points raised during the review process.

We look forward to receiving your revised manuscript.

Kind regards,

Muhammad Athar, PhD

Academic Editor

PLOS ONE

Journal Requirements:

Reviewers' comments:

Reviewer's Responses to Questions

**Comments to the Author**

1. Is the manuscript technically sound, and do the data support the conclusions?

Reviewer #1: Partly

Reviewer #2: Yes

2. Has the statistical analysis been performed appropriately and rigorously?

Reviewer #1: Yes

Reviewer #2: Yes

3. Have the authors made all data underlying the findings in their manuscript fully available?

Reviewer #1: No

Reviewer #2: Yes

4. Is the manuscript presented in an intelligible fashion and written in standard English?

Reviewer #1: Yes

Reviewer #2: Yes

Reviewer #1: Page 3, line 84. This dataset seems to be a population based registry, in the sense that all fatalities occurred in the Country are included, But it is not a trauma registry and many information regarding injuries are lacking. There is no selection of major trauma and also minor injuries are included. The only stratification of severity is fatal and non-fatal crash. If this consideration is correct these concepts should be stressed in the text

Page 4, line 122. You cannot define with available data injury severity, but only injury fatality

Page 7, line 176-187. Data reported in the text are presented in table 2. The text should comment the table without repeating all the data, outlining more important results

In the results the weight of injuries of different body region on fatal outcome has been explored in table 2, but only regarding head and neck injuries. It has been showed that chest injuries are important determinant of death in older motorcyclists. From your data is it possible to derive further information about injuries more associated with fatalities? Limitations of the study are addressed, but if injured regions are available some discussion can be added. In general, the paper is interesting with an important population. The limit is that many sanitary data are lacking and results are more oriented toward problems of limited interest for doctors and nurses

Reviewer #2: Effects of aging and risk-taking behaviors on fatal injuries among old motorcyclists in Taiwan: evidence from 2011 to 2022

1. Overall Assessment

This is a well-structured and important epidemiological study examining the joint effects of aging and risk-taking behaviors (unhelmeted, unlicensed, and drunk riding) on fatal motorcycle injuries among the elderly in Taiwan. The study leverages a large national dataset (n = 338,231), applies sound statistical methods, and provides policy-relevant conclusions. It fills an important gap by specifically targeting very old motorcyclists (≥75 years) and interaction effects with risk-taking behavior.

However, several aspects require clarification, deeper analysis, and improved presentation to strengthen the study’s rigor, clarity, and contribution.

2. Strengths

• Large sample size from a reliable national database enhances generalizability.

• Clear definition of variables and outcome measures.

• Use of joint effects models and subgroup analysis is commendable.

• Public health and policy relevance is high.

• Ethical clarification and transparency about data usage are adequately addressed.

3. Issues

a) Clarification of Study Design and Ethics

• While the authors corrected the date of data access (from 2023 to 2024), proof of prospective ethical approval before data access must be clear to meet journal standards. Consider appending a dated ethics approval letter.

b) Causality Interpretation

• While the authors rightly avoid overclaiming causality, the discussion sometimes implies causal links (e.g., "these behaviors were contributors to fatal injuries"). Reword to emphasize association rather than causality.

c) Missing Confounders

• Some potentially important confounders are not considered:

o Vehicle type/power (e.g., electric vs. gas motorbike).

o Road condition complexity (e.g., curves, intersections).

o Rider comorbidities or medication use, though these may not be available.

o Crash partner behavior or demographics—acknowledged as a limitation but deserves stronger discussion.

d) Terminology

• Avoid using “very old motorcycles” in the abstract (line 32). It should be “very old motorcyclists.”

• Use consistent age group terminology: sometimes written as “≥75 years,” sometimes “very old.” Define this clearly early on and use consistently.

e) Language and Style

• Some phrasing is awkward or grammatically incorrect. For instance:

o “This would underscore the importance of tailored interventions among individuals with a combination of 75 years or above and unlicensed riding…” – should be revised for clarity.

o “This particular group of motorcyclists are prohibited from riding if they not fitted enough” – should be “...if they are not fit enough.”

Suggest proofreading by a native English editor (beyond Wallace Editing) or using a language enhancement tool.

c) Figures

• Figure 2 and 3 forest plots are mentioned but not included in the PDF (only URLs).

o Reviewers need figures embedded or annexed for full appraisal.

o Please include sample sizes per subgroup in forest plots.

5. Conclusion and Recommendation

This manuscript addresses a timely and critical road safety issue using robust data and appropriate methods. The focus on very old motorcyclists and interaction with behavioral risks is novel. However, moderate revisions are required to improve clarity, statistical transparency, and precision of interpretation.

Recommendation: Minor Revision

**Do you want your identity to be public for this peer review?** For information about this choice, including consent withdrawal, please see our Privacy Policy

Reviewer #1: No

Reviewer #2: No

---

## [Author Response · Author response to Decision Letter 1]

13 Jun 2025

Dear Editors and Reviewers,

We greatly appreciate the valuable comments and suggestions raised by reviewers. Please very kindly see our responses below, as well as the revised manuscript. We would be glad if you could have our manuscript reviewed again.

Best regards

Chih-Wei Pai (Prof)

Graduate Institute of Injury Prevention and Control College of Public Health, Taipei Medical University

Reviewer 1 Comments

1. Page 3, line 84. This dataset seems to be a populationbased registry, in the sense that all fatalities occurred in the Country are included, But it is not a trauma registry and many information regarding injuries are lacking. There is no selection of major trauma and also minor injuries are included. The only stratification of severity is fatal and non-fatal crash. If this consideration is correct these concepts should be stressed in the text.

Author’s response: We greatly appreciate this reviewer’s comment. In our dataset, detailed clinical information such as Injury Severity Scores (ISS) or Abbreviated Injury Scale (AIS) are not available. However, in our manuscript, we followed a widely accepted and previously published classification using this dataset, whereby we categorized injury severity into two groups: fatal and non-fatal injuries. Fatal injury was defined as deaths resulting from crashes, while non-fatal injuries included all other recorded injury severities except for fatalities. This classification approach has been consistently applied in prior peer-reviewed studies using the same dataset, including:

• Wiratama, B. S., Chen, P. L., Ma, S. T., Chen, Y. H., Saleh, W., Lin, H. A., & Pai, C. W. (2020). Evaluating the combined effect of alcohol-involved and un-helmeted riding on motorcyclist fatalities in Taiwan. Accident; analysis and prevention, 143, 105594. https://doi.org/10.1016/j.aap.2020.105594

• Lin, H. A., Chan, C. W., Wiratama, B. S., Chen, P. L., Wang, M. H., Chao, C. J., Saleh, W., Huang, H. C., & Pai, C. W. (2022). Evaluating the effect of drunk driving on fatal injuries among vulnerable road users in Taiwan: a population-based study. BMC public health, 22(1), 2059. https://doi.org/10.1186/s12889-022-14402-3

Therefore, to address your suggestion, we have revised the Methods section (Page 5, line 91) as follows:

“In our dataset, detailed clinical information such as Injury Severity Scores (ISS) or Abbreviated Injury Scale (AIS) are not available. Therefore, injury fatality is categorized as fatal or nonfatal. Fatal injury was defined as a case in which the riders died following crashes, whereas a non-fatal injury included any other recorded injury category aside from death.”

2. Page 4, line 122. You cannot define with available data injury severity, but only injury fatality

Author’s response: We greatly appreciate this reviewer’s comment. In response, we have revised the term “injury severity” to “injury fatality” throughout the manuscript (Line 29, 90, 92, 96, 99, 125, 127, 172, 259).

3. Page 7, line 176-187. Data reported in the text are presented in table 2. The text should comment the table without repeating all the data, outlining more important results.

Author’s response: We appreciate the reviewer’s comment and understand the concern regarding redundancy between the text and Table 2. Therefore, we revised our manuscript (Page 11, line 183) as follows:

“Additional risk factors for fatal injuries included risk-taking behaviors such as unhelmeted riding (AOR = 3.581; CI = 3.171–4.040), unlicensed riding (AOR = 1.382; CI = 1.259–1.517), and drunk riding (AOR = 3.861; CI = 3.266–4.564).”

4. In the results the weight of injuries of different body region on fatal outcome has been explored in table 2, but only regarding head and neck injuries. It has been showed that chest injuries are important determinant of death in older motorcyclists. From your data is it possible to derive further information about injuries more associated with fatalities? Limitations of the study are addressed, but if injured regions are available some discussion can be added. In general, the paper is interesting with an important population. The limit is that many sanitary data are lacking and results are more oriented toward problems of limited interest for doctors and nurses.

Author’s response: We sincerely thank the reviewer for this thoughtful comment. As noted, Table 1 of our manuscript presents the distribution of injuries by body region, including head and neck, extremities, chest/abdomen, spine, and unknown. In Table 2, we include only head and neck injuries in the model because this injury category accounted for the highest proportion of fatalities among all body regions, indicating a particularly strong association with fatal outcomes in our dataset. Including only the most impactful injury region was also aimed at minimizing model complexity.

We fully acknowledge the reviewer’s important point regarding the clinical relevance of chest and abdominal injuries. While our current analysis focuses specifically on fatalities and head and neck injuries, we will aim to expand our analysis in the future studies by including other critical injury regions such as the chest and abdominal injuries which could enhance the clinical applicability of our findings. We believe this will enhance the clinical applicability of our findings and provide greater relevance for medical professionals, including physicians, nurses, and trauma surgeons.

Reviewer 2 Comments

1. This is a well-structured and important epidemiological study examining the joint effects of aging and risk-taking behaviors (unhelmeted, unlicensed, and drunk riding) on fatal motorcycle injuries among the elderly in Taiwan. The study leverages a large national dataset (n = 338,231), applies sound statistical methods, and provides policy-relevant conclusions. It fills an important gap by specifically targeting very old motorcyclists (≥75 years) and interaction effects with risk-taking behavior. However, several aspects require clarification, deeper analysis, and improved presentation to strengthen the study’s rigor, clarity, and contribution.

Author’s response: We greatly appreciate this reviewer’s comment.

2. Strengths

2.1 Large sample size from a reliable national database enhances generalizability.

2.2 Clear definition of variables and outcome measures.

2.3 Use of joint effects models and subgroup analysis is commendable.

2.4 Public health and policy relevance is high.

2.5 Ethical clarification and transparency about data usage are adequately addressed.

Author’s response: We greatly appreciate this reviewer’s comment.

3. Issues

3.1 Clarification of Study Design and Ethics. While the authors corrected the date of data access (from 2023 to 2024), proof of prospective ethical approval before data access must be clear to meet journal standards. Consider appending a dated ethics approval letter.

Author’s response: We greatly appreciate the reviewer’s comment. To ensure clarity and confirm that ethical approval was obtained prior to study initiation, we have revised the statement to include the approval date. The updated sentence now reads (Page 17, Line 295):

“This study was conducted in accordance with the Declaration of Helsinki principles and approved by the Research Ethics Review Committee, New Taipei City Hospital (113001-E) on March 4th, 2024.”

In addition, we have attached the IRB approval for your reference.

3.2 Causality Interpretation. While the authors rightly avoid overclaiming causality, the discussion sometimes implies causal links (e.g., "these behaviors were contributors to fatal injuries"). Reword to emphasize association rather than causality.

Author’s response: We greatly appreciate the reviewer’s comment. In response, we have revised several sentences in our manuscript, replacing phrases such as "contributors" and "due to" with more appropriate terms such as "associated with" and "possibly attributable to" to avoid overclaiming causality. The revised version now reads:

“Our findings elucidated that those risk-taking behaviors, including unhelmeted riding, unlicensed riding, and drunk riding, were associated with fatal injuries.” (Page 14, line 218).

“Fatality rates of motor vehicle crashes among the old population have risen, primarily in association with age-related declines in health and functional abilities.” (Page 2, line 23).

“Moreover, unhelmeted riding was associated with higher fatality rates among old motorcyclists, possibly attributable to diminished resilience and decline in physical fitness.” (Page 14, line 226).

3.3 Missing Confounders. Some potentially important confounders are not considered: Vehicle type/power (e.g., electric vs. gas motorbike), road condition complexity (e.g., curves, intersections), rider comorbidities or medication use, though these may not be available, and crash partner behavior or demographics. Acknowledged as a limitation but deserves stronger discussion.

Author’s response: We greatly appreciate the reviewer’s insightful comments. We acknowledge that factors such as vehicle type/power (e.g., electric vs. gas motorbikes), road geometry (e.g., curves and intersections), rider comorbidities or medication use, and crash partner characteristics may influence crash outcomes. However, our current dataset only includes gas-powered motorcycles and lacks distinction between electric and gas motorbikes. Detailed information on road geometry is also unavailable in the national crash database; we attempted to address road conditions through the available "road surface condition" variable (dry or slippery). While we are exploring the use of GIS and supplementary sources (e.g., Google Maps) for road geometry, integrating such data remains beyond the scope of this study due to limited public access. Similarly, data on crash partner behavior and demographics, as well as rider comorbidities or medication use, are not captured in the dataset and would require linkage with clinical records or custom-designed data collection. We acknowledge these limitations and recommend that future studies incorporate such variables to strengthen risk assessments and prevention strategies.

We recognize the importance of this perspective and consider it a valuable direction for future research. We will take reviewer’s recommendations and put in our limitation section as follows (Page 15, line 255):

“These characteristics are important, as they can significantly affect crash outcomes. For instance, motorcycle riders were more likely to sustain fatal injuries when they were struck by unlicensed, unrestrained, and drunk car drivers [29,47].”

3.4 Terminology. Avoid using “very old motorcycles” in the abstract (line 32). It should be “very old motorcyclists”. Use consistent age group terminology: sometimes written as “≥75 years,” sometimes “very old.” Define this clearly early on and use consistently.

Author’s response: We greatly appreciate the reviewer’s comment. We have revised the phrase “very old motorcycles” to “very old motorcyclists” in the abstract (Page 2, Line 33). In addition, we would like to clarify that we defined “very old motorcyclists” for those aged 75 years or above early in the manuscript (Page 2, Line 33). To maintain consistency and clarity throughout the paper, we primarily used the term “very old motorcyclists” after this definition. Additionally, in the statistical analysis section (Page 7, Line 133), we provided a detailed explanation of the joint effect model, explicitly stating the comparison between motorcyclists aged 75 years or above and their younger counterparts.

3.5 Language and Style. Some phrasing is awkward or grammatically incorrect. For instance: “This would underscore the importance of tailored interventions among individuals with a combination of 75 years or above and unlicensed riding…” – should be revised for clarity. “This particular group of motorcyclists are prohibited from riding if they not fitted enough” – should be “...if they are not fit enough.”. Suggest proofreading by a native English editor (beyond Wallace Editing) or using a language enhancement tool.

Author’s response: We greatly appreciate the reviewer’s comment. We have revised the sentence for clarity and grammatical accuracy (Page 3, line 48). In addition, we also revised in statistical analysis section (Page 8, line 154). The revised version now reads:

“This underscores the importance of tailored interventions targeting individuals aged 75 years or above who engage in unlicensed riding, which remains an important area for continued research.” (Page 3, line 48).

“This particular group of motorcyclists is prohibited from riding if they are not fit enough.” (Page 8, line 154).

3.6 Figures. Figure 2 and 3 forest plots are mentioned but not included in the PDF (only URLs). Reviewers need figures embedded or annexed for full appraisal. Please include sample sizes per subgroup in forest plots.

Author’s response: We greatly appreciate the reviewer’s comment. We sincerely apologize for the inconvenience caused by the absence of Figures 2 and 3 in the PDF version. We would like to clarify that Figures 2 and 3 were submitted separately as high-resolution files during the journal submission process, in accordance with the journal’s formatting guidelines, which require figures to be removed from the main manuscript. Additionally, to ensure accessibility, we included a secure online link within the manuscript so that readers and reviewers can conveniently view the figures when needed. In response to your suggestion, we have updated the forest plots to include the sample sizes for each subgroup, and the revised figures have been re-submitted accordingly.

---

## [Decision Letter · Decision Letter 1]

8 Jul 2025

Effects of aging and risk-taking behaviors on fatal injuries among old motorcyclists in Taiwan: evidence from 2011 to 2022

PONE-D-25-14406R1

Dear Dr. Pai,

We’re pleased to inform you that your manuscript has been judged scientifically suitable for publication and will be formally accepted for publication once it meets all outstanding technical requirements.

Kind regards,

Muhammad Athar, PhD

Academic Editor

PLOS ONE

Additional Editor Comments (optional):

Reviewers' comments:

Reviewer's Responses to Questions

**Comments to the Author**

Reviewer #1: All comments have been addressed

Reviewer #2: All comments have been addressed

2. Is the manuscript technically sound, and do the data support the conclusions?

Reviewer #1: Yes

Reviewer #2: Yes

3. Has the statistical analysis been performed appropriately and rigorously?

Reviewer #1: Yes

Reviewer #2: Yes

4. Have the authors made all data underlying the findings in their manuscript fully available?

Reviewer #1: Yes

Reviewer #2: Yes

5. Is the manuscript presented in an intelligible fashion and written in standard English?

Reviewer #1: Yes

Reviewer #2: Yes

Reviewer #1: All the questions have been addresses and I have no other observations for the manuscript. Although data which were presented and discussed derive from a non-sanitary data base, some interesting information can be obtained

Reviewer #2: (No Response)

**Do you want your identity to be public for this peer review?** For information about this choice, including consent withdrawal, please see our Privacy Policy

Reviewer #1: No

Reviewer #2: **Yes: ** Noor Diana Binti Abdul Majid

---

## [Editor Report · Acceptance letter]

PONE-D-25-14406R1

PLOS ONE

Dear Dr. Pai,

I'm pleased to inform you that your manuscript has been deemed suitable for publication in PLOS ONE. Congratulations! Your manuscript is now being handed over to our production team.

Kind regards,

on behalf of

Dr. Muhammad Athar

Academic Editor

PLOS ONE